# The role of heterochronic gene expression and regulatory architecture in early developmental divergence

Nathan D Harry, Christina Zakas*

Department of Biological Sciences, North Carolina State University, Raleigh, United States

**Abstract** New developmental programs can evolve through adaptive changes to gene expression. The annelid *Streblospio benedicti* has a developmental dimorphism, which provides a unique intraspecific framework for understanding the earliest genetic changes that take place during developmental divergence. Using comparative RNAseq through ontogeny, we find that only a small proportion of genes are differentially expressed at any time, despite major differences in larval development and life history. These genes shift expression profiles across morphs by either turning off any expression in one morph or changing the timing or amount of gene expression. We directly connect the contributions of these mechanisms to differences in developmental processes. We examine $F_1$ offspring – using reciprocal crosses – to determine maternal mRNA inheritance and the regulatory architecture of gene expression. These results highlight the importance of both novel gene expression and heterochronic shifts in developmental evolution, as well as the *trans*-acting regulatory factors in initiating divergence.

**\*For correspondence:**
czakas@ncsu.edu

**Competing interest:** The authors declare that no competing interests exist.

## eLife assessment

This **important** study examines the extent to which distinct developmental pathways that result in alternative morphs correlate with transcriptome differences in a marine annelid, Streblospio benedicti. The strengths of the study include the experimental design and dense temporal sampling, which together provide **convincing** evidence that the two morphs can be clearly distinguished at the transcriptome level, despite relatively modest overall differences. The work will be of particular interest to students of the evolution of development.

## Introduction

Small changes in development can result in vast morphological differentiation and divergence. Through the history of evolutionary developmental biology, researchers have proposed that changes in the timing of traits during development can produce most morphological changes (*deBeer, 1930*; *Gould, 1985*; *Dobreva et al., 2022*). Though this theory has been further refined in the field since Haeckel first coined the term 'heterochrony' in development (1875), the timing of developmental changes remains a prominent mechanism of diversification (*Wray and McClay, 1989*; *Wray and Raff, 1991*; *Smith, 2002*; *Smith, 2003*). At the molecular level, changes in gene expression timing (**heterochronic** genes) or expression amount (**heteromorphic** genes) can underlie major morphological differences (*Erwin and Davidson, 2002*). However, these heterochronic shifts are usually investigated on a per-gene basis to reveal the extent that morphological change is achieved through gene expression perturbation (as reviewed in *Dobreva et al., 2022*). Overall, the extent that heterochronic gene expression contributes to developmental differences at the molecular level has not been quantified.

Therefore, the underlying regulatory changes that result in gene expression changes also remain elusive. In this study we quantify total gene expression over sequential developmental time to determine the extent that gene expression differences occur between divergent developmental and life-history modes. Furthermore, we use genetic crosses between the developmental morphs to quantify the mode of regulatory change that is responsible for these differences.

Changes in gene expression (heterochrony and heteromorphy) are now well established drivers of both interspecific (*King and Wilson, 1975*; *Carroll, 1995*; *Wray et al., 2003*; *Fay and Wittkopp, 2008*) and intraspecific (*López-Maury et al., 2008*; *Hamann et al., 2021*) differentiation. Knowing the extent that these biological processes are at play is fundamental to understanding developmental evolution (*Gould, 1985*; *Raff and Wray, 1989*; *Smith, 2003*; *Vaglia and Smith, 2003*; *Carleton et al., 2008*; *Gunter et al., 2014*; *Willink et al., 2020*). While somewhat subtle differences in gene expression timing and amount can drive morphological changes, *morph-specific* genes – where genes are simply turned on or off, and possibly gained or lost in one lineage – are also possible (*Hilgers et al., 2018*; *Luna and Chain, 2021*). Despite the importance of these mechanisms for developmental and evolutionary change, long-standing questions in the field remain: What are the extents to which modifications of gene expression change developmental programs and morphological outcomes? What are the molecular factors that regulate these changes and how do they act?

One difficulty in determining the genetic basis of gene expression shifts is that it typically involves parsing developmental programs across divergent taxa, and therefore requires querying numerous genetic differences that could have arisen by either selection or drift over millions of years. Creating hybrids across divergent species presents its own difficulties. Here, we determine the extent that gene expression differences contribute to developmental divergence over small evolutionary timescales using a model with two developmental modes within a single species. We examine the prevailing prediction that heterochrony is the primary driver of morphological change and developmental evolution (*Raff and Wray, 1989*; *McNamara, 2012*; *Dobreva et al., 2022*). While numerous interspecific gene expression studies have assessed the occurrence of both heterochronic genes and morph-specific genes (*Ruvkun and Giusto, 1989*; *Capra et al., 2010*; *Simola et al., 2010*; *Schmitz et al., 2020*), no study has assessed the contributions of these mechanisms to producing differences in developmental processes. By using an intraspecific model of developmental divergence where genetic crosses are possible, we determine the architecture of regulatory differences that drive gene expression and ultimately life-history differences.

## Results

### Study system and embryology

We compare the gene expression over developmental time for two morphs of a marine annelid, *Streblospio benedicti*, which has an intraspecific developmental dimorphism. There are two distinct developmental morphs which differ in their egg size, embryological development time, larval ecology, and morphology. These are either obligately feeding *planktotrophic* (PP) larvae or non-feeding *lecithotrophic* (LL) larvae. Despite these developmental differences, as adults they are morphologically indistinguishable outside of some reproductive traits and occupy the same environmental niches. The larval traits are heritable, meaning the differences in development are genetic and not plastic (*Levin and Creed, 1986*). The two morphs have been characterized extensively in terms of life-history and genetic differences (*Levin, 1984*; *Levin and Creed, 1986*; *Gibson et al., 2010*; *Zakas and Rockman, 2014*; *Zakas et al., 2018*; *Zakas, 2022*). Intriguingly, crosses between the morphs are viable with no obvious fitness effects, and these $F_1$ offspring can have intermediate larval traits compared to the parentals (*Levin and Creed, 1986*; *Zakas, 2022*). Embryological differences between the two types (and thus egg sizes) have been briefly described (*McCAIN, 2008*) but here we detail the full time course of embryogenesis from the one-cell embryo through the larval phase in detail for both morphs, and present data from reciprocal $F_1$ crosses (PL or LP) between the morphs in both directions across the same developmental period.

Spiralian animals have a famously conserved pattern of embryological cleavage (reviewed in *Lyons et al., 2012*), so unsurprisingly we find the two morphs proceed through the same developmental stages despite starting from eggs of different sizes (8× volume difference). Despite the similarity in embryo morphology, there are some notable differences between the two morphs: the absolute

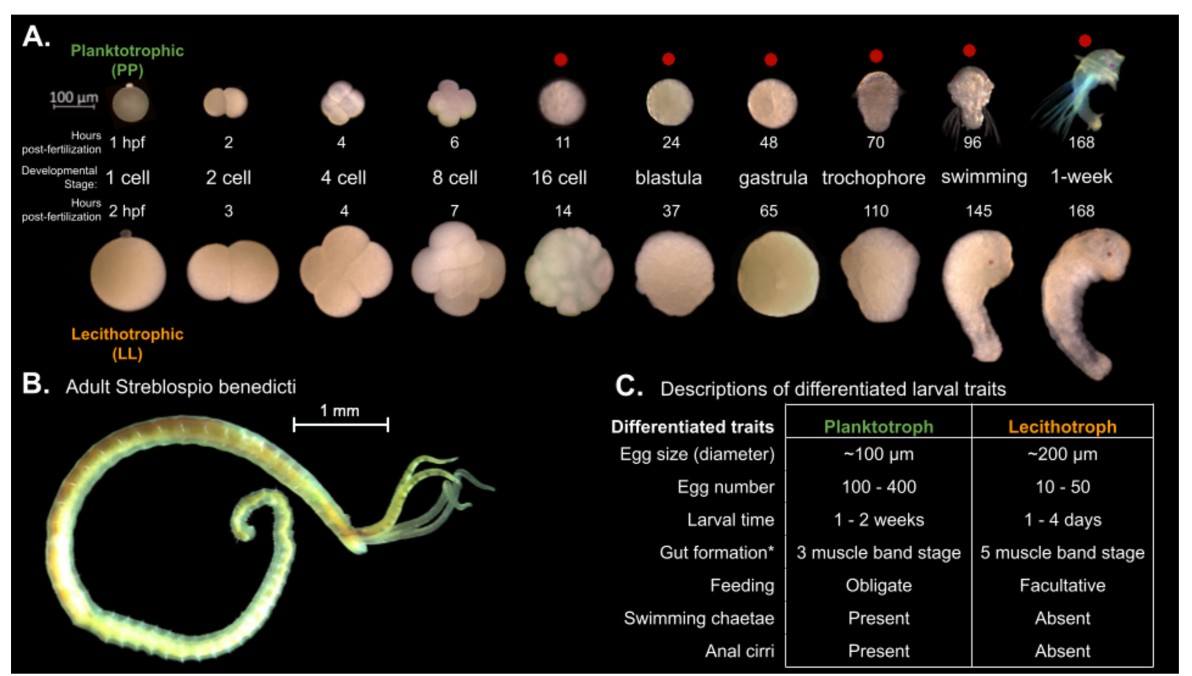

**Figure 1.** Embryology and development of *S. benedicti*. (**A**) Timeline of early development of the two morphs. The 'swimming' stage is the onset of swimming ability in both morphs. The '1-week' old stage (7 days post fertilization) occurs approximately 1–4 days following larval release from maternal brood pouches under natural conditions. Planktotrophic (PP) larvae at this stage were not fed. Stages used in this study are marked with a red dot. (**B**) Adult male. The adults of both morphs are indistinguishable outside of reproductive traits. (**C**) Table of larval traits. Lecithotrophic (LL) offspring are facultatively feeding, meaning that feeding is not obligatory, but they are capable of feeding (*Pernet and McArthur, 2006*). Gut formation from *Pernet and McHugh, 2010*, where the three-band stage is our swimming stage and the five-band stage is shortly before our 1-week stage.

time between each stage is shifted such that the LL embryos take longer to reach an equivalent larval stage, which is expected as larger, more yolky cells can take longer to divide (*McCAIN, 2008*). At the swimming larval stage there are some notable morphological and behavioral differences (reviewed in *Zakas, 2022*, *Figure 1*), notably the PP larvae are obligately feeding and have feeding structures, while the LL are not. While the embryological stages may not be morphologically different other than size, it is reasonable to expect that they could be expressing different genes or changing the relative timing of expression to produce the morphological and behavioral differences seen in the larvae.

## Total RNA expression analysis

We measured total gene expression from the six developmental stages using RNAseq (*Figure 2*) with at least four biological replicates per morph at each stage (*Figure 2A*). Using the full dataset, the first two principal component analysis (PCA) show most of the variance in total gene expression is due to developmental stage and morph (PC1; *Figure 2B*). As expected the LL individuals appear to transcriptionally fall behind the PP offspring at each morphologically defined, equivalent developmental stage. We expect this pattern as LL offspring develop more slowly to reach these first developmental stages but reach the juvenile stage more quickly in absolute time than PP offspring, which develop in the water column for a longer period (*Figure 1*). Notably, pre-gastrulation development is distinctly separated from post-gastrulation by the second principal component.

As expected for an intraspecific comparison, most genes are conserved, having the same expression levels at all stages in both morphs. Only 36.2% of all expressed genes are significantly differentially expressed (DE) between PP and LL at any stage in this dataset. We find that early in development over a third of these DE genes are significantly different between the morphs, but these differences tend to be quite small in magnitude. At gastrulation the number of significant genes decreases to less than 5% of the total DE genes, however these remaining expression differences are much larger in magnitude (*Figure 2C*). It appears that the two morphs are more functionally distinct during early

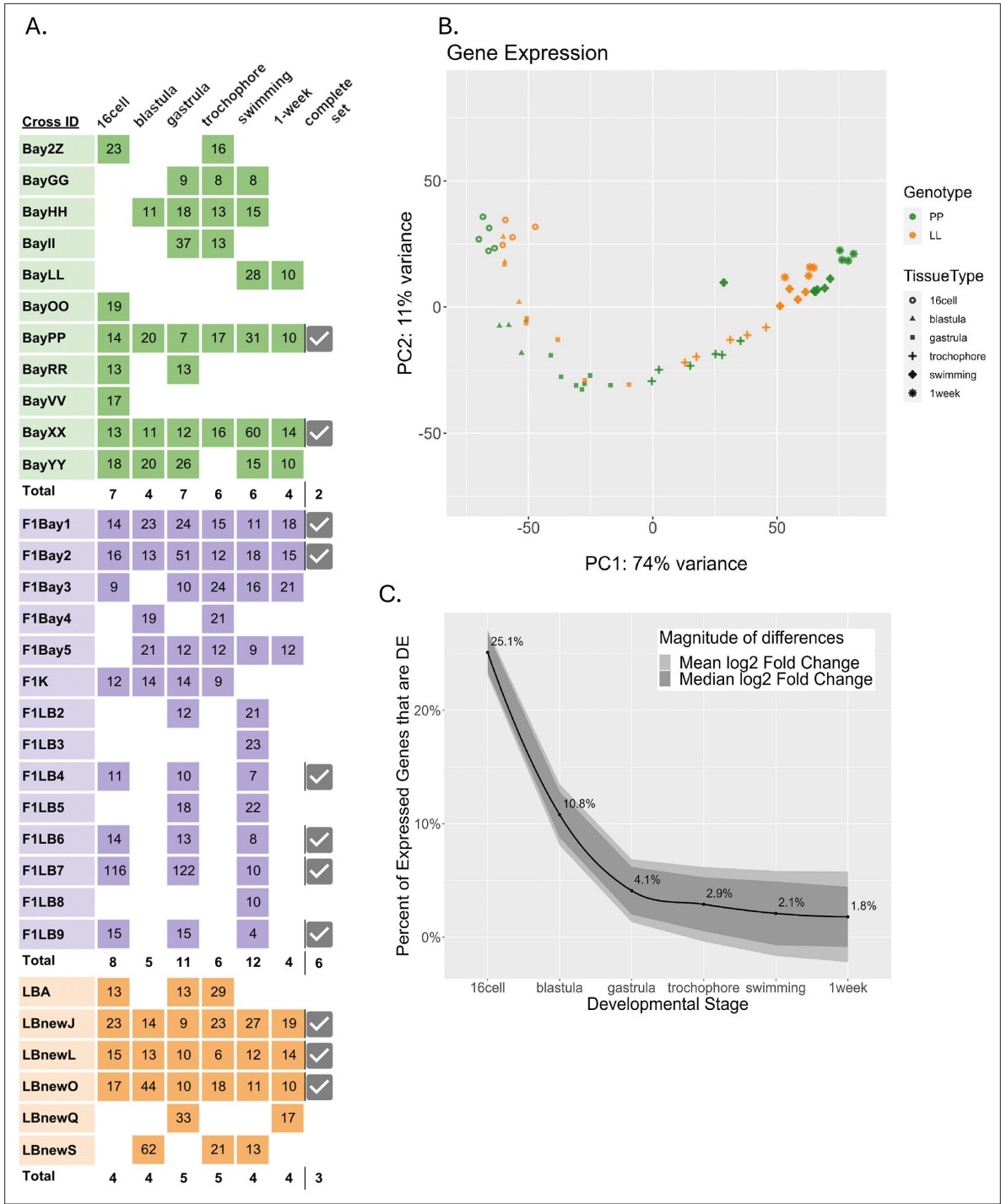

**Figure 2.** Differentially expressed (DE) genes across six developmental stages in the two developmental morphs. (**A**) Sequenced libraries. Colors indicate morphs: green = PP (planktotrophic), orange = LL (lecithotrophic), purple = (F₁) PL, pink = (F₁) LP. Numbers in boxes are the sample's sequencing depth in million reads. Total number of replicates for each time point are in bold. Samples for which all six stages were collected from the same cross are indicated as 'complete' with a check mark. (**B**) Principal component analysis of PP and LL samples. PC1 represents developmental time, whereas PC2 separates pre-gastrulation from post-gastrulation (**C**) DE between morphs over development. Black line is the percentage of total expressed genes that are DE at each stage, ribbon displays the relative mean and median log2 fold-change between morphs for those genes that are DE.

The online version of this article includes the following figure supplement(s) for figure 2:

**Figure supplement 1.** Read mapping rate between samples.

**Figure supplement 2.** Number of genes expressed at each stage in planktotrophic (PP) and lecithotrophic (LL) samples.

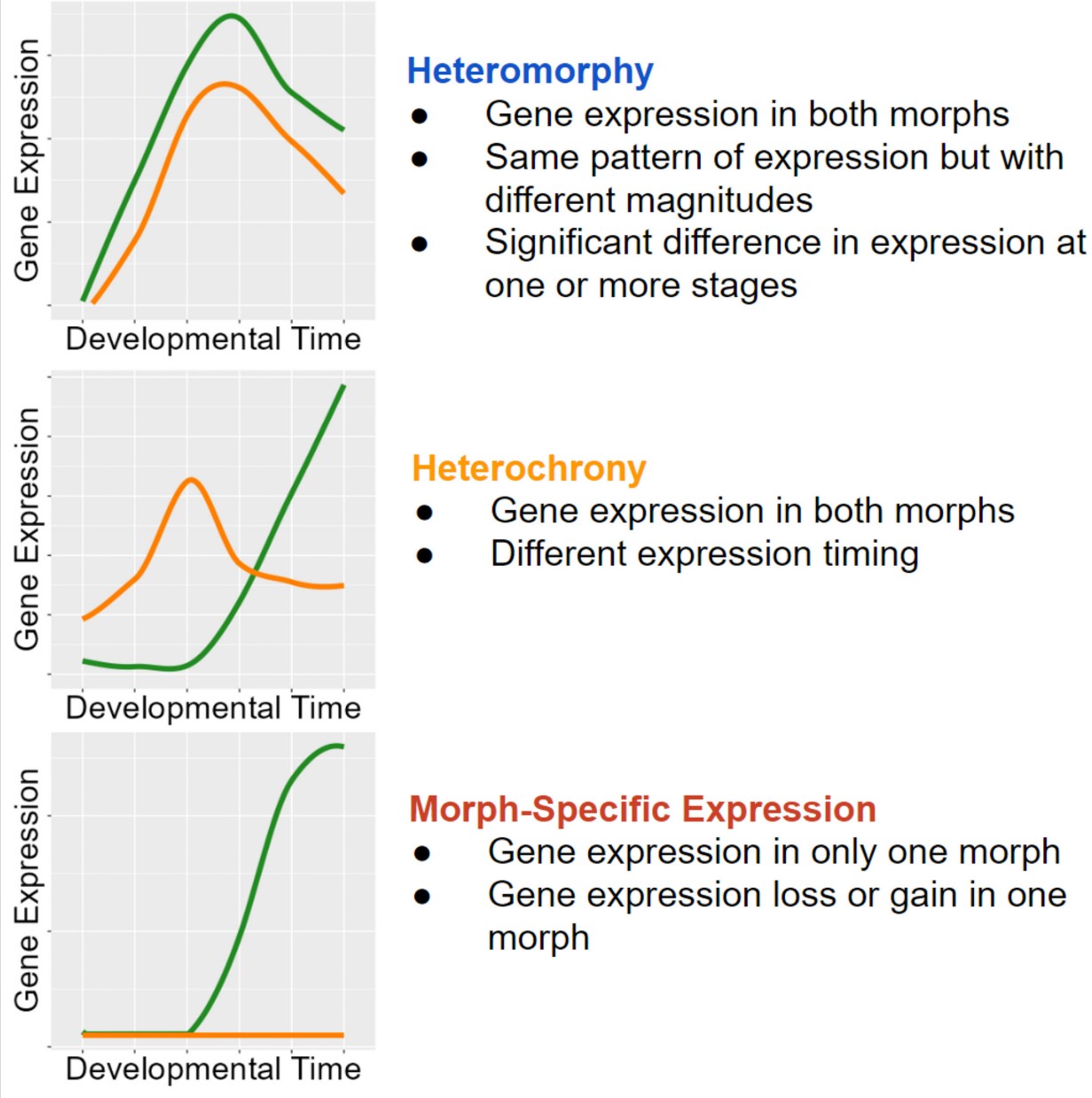

**Heteromorphy**
- Gene expression in both morphs
- Same pattern of expression but with different magnitudes
- Significant difference in expression at one or more stages

**Heterochrony**
- Gene expression in both morphs
- Different expression timing

**Morph-Specific Expression**
- Gene expression in only one morph
- Gene expression loss or gain in one morph

**Figure 3.** Classifications with examples of gene expression categories. Functional heterochrony is a change in the developmental stage at which a particular pattern of gene expression appears instead of a difference in absolute time.

development, likely because of the different metabolic requirements imposed on them by the differences in maternal egg provisioning (*Moran and McAlister, 2009*; *Zakas, 2022*; *Harry and Zakas, 2023*).

## Differential expression analysis

We use our RNAseq time course dataset to quantify the extent that modifications to gene expression timing and amount contribute to developmental differences. We define **heteromorphic** genes as homologous genes whose expression differs significantly between larval morphs, but the pattern of expression (expression profile) does not change (*Figure 3*). The significant difference in heteromorphic genes could be at only one or a few discrete time points. **Heterochronic** genes are those whose overall expression pattern and timing change between the developmental morphs, as discussed in the clustering algorithm below. We categorize genes that are only expressed in one morph over the

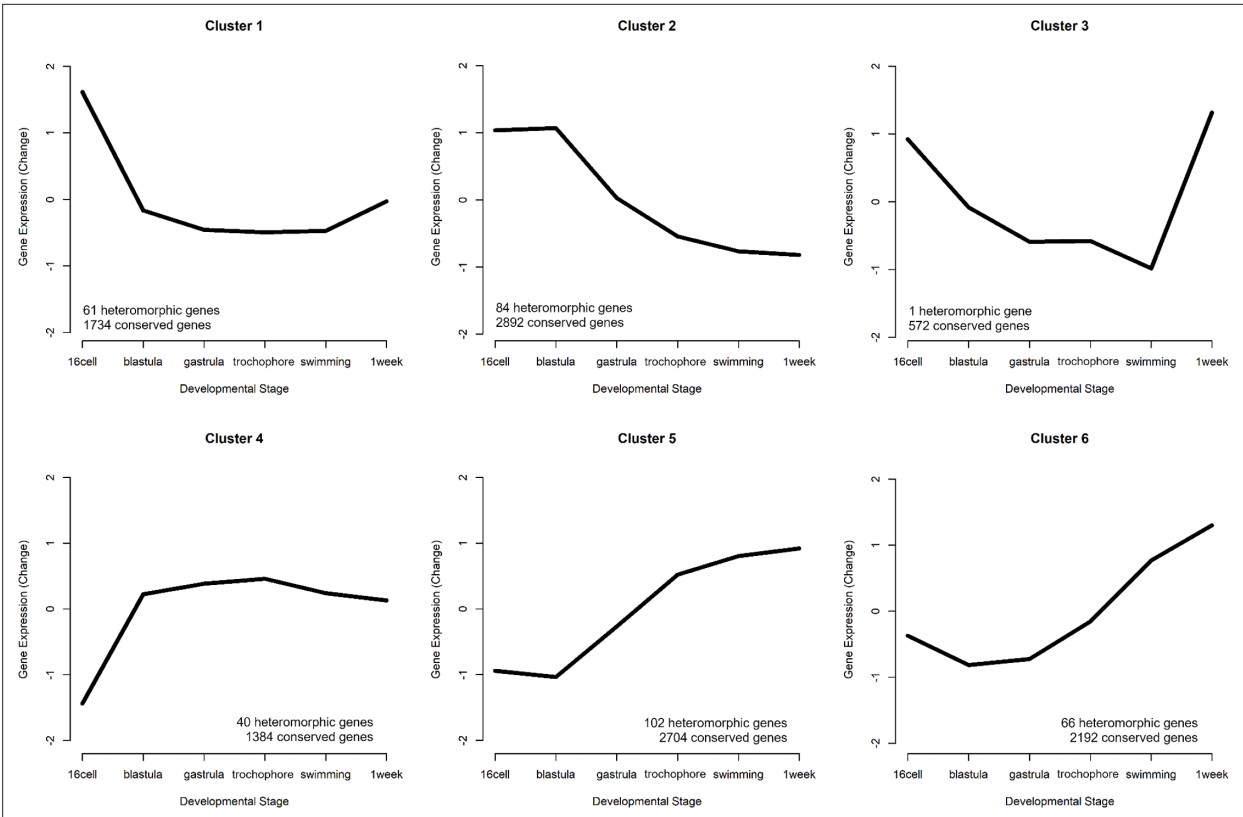

**Figure 4.** Six gene expression profiles representing the patterns of expression. The number of conserved genes and heteromorphic genes that match each cluster is listed to summarize overall trends in expression patterns that appear mostly similar in both morphs. Clusters 2 (early expression) and 5 (late expression) represent the most genes.

The online version of this article includes the following figure supplement(s) for figure 4:

**Figure supplement 1.** Cluster number optimization.

**Figure supplement 2.** Clusters with all genes shown plotted on top of cluster core genes.

full time course of development as **morph-specific** genes. These genes are specific to only one developmental type.

To differentiate heterochronic shifts in gene expression from heteromorphic ones, we clustered all gene expression patterns from PP into a representative set of expression profiles using Mfuzz (v2.60.0) (no additional clusters were found when using LL). We selected six clusters based on the criteria that the average correlation between cluster pairs increased with each additional cluster, such that the final clusters generated sufficiently represented the diversity of gene expression patterns (*Figure 4*, *Figure 4—figure supplement 1*). We then assigned each gene to a cluster in the PP and LL datasets independently. A gene's expression is considered heterochronic when it appears in one cluster for PP and a different cluster for LL.

We identified 354 genes from our set of DE genes (45.9%) where both the PP and LL expression pattern matched the same cluster. (These are heteromorphic genes and are significantly DE in at least one developmental stage, but do not have different profiles of expression between PP and LL.) Approximately half of these are assigned to clusters 2 and 5. Cluster 2, which shows a pattern of maternal transcript degradation with no subsequent zygotic expression, is likely to contain genes associated with embryogenesis that are shared by both morphs. Cluster 5 shows a pattern of largely post-gastrulation zygotic gene expression. Based on this pattern, these genes are likely to be associated with shared larval features. Although there are some late-appearing heteromorphic genes that are associated with discrete larval differences: 'chitin catabolic processes', for example, are overexpressed in PP at the swimming stage (when they grow swimming chaetae) and are lowly expressed in LL which do not make swimming chaetae (see *Supplementary file 1*).

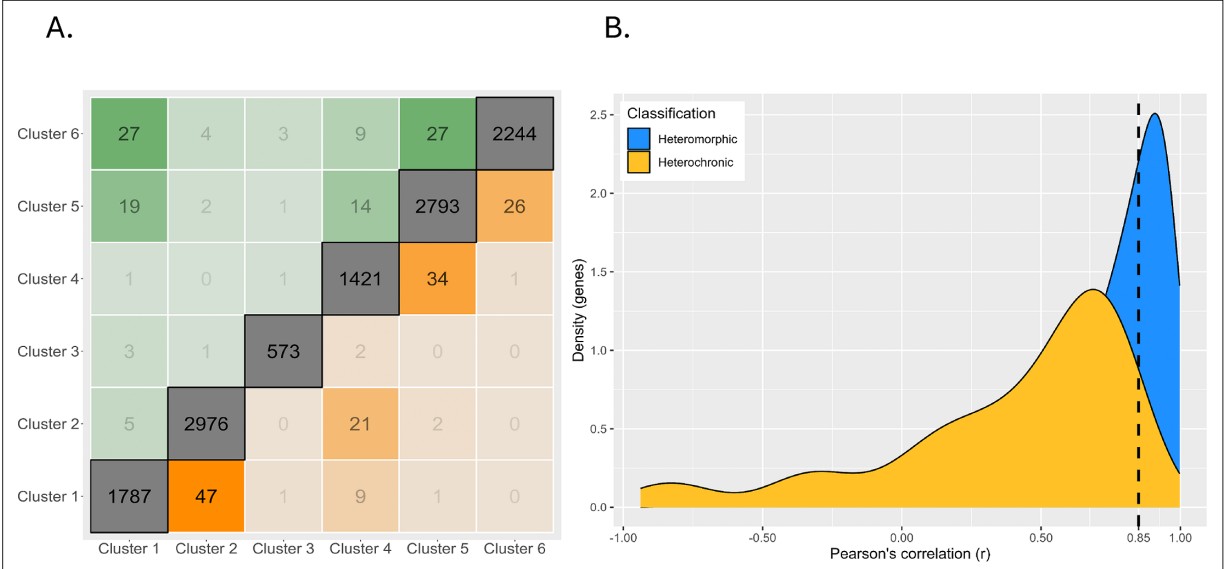

**Figure 5.** Magnitude of expression profile changes. (**A**) Number of switches between each cluster. Clusters are arranged on the axes in order from high early developmental expression (left/bottom) to high late developmental gene expression (right/top). The diagonal represents genes which are assigned to the same cluster in both planktotrophic (PP) and lecithotrophic (LL) samples, boxes near the diagonal are switches between similar cluster profiles, and boxes furthest from the diagonal are extreme cluster switches. Genes expressed earlier in PP are above diagonal and earlier in LL are below. (**B**) Density plot of genes correlations between PP and LL. Heterochronic genes are yellow and heteromorphic genes in blue. Most heterochronies are minor changes in gene expression timing but the left-tailed distribution shows that few genes have large differences in timing.

Next, we identified heterochronies, which we define as a gene that switches clusters – and therefore expression profiles – between PP and LL. There are 224 heterochronic genes, which is 29% of DE genes. As we reduced the complexity of the entire gene expression dataset into six clusters, some genes with similar profiles in both PP and LL were ultimately assigned to different clusters, generating a false positive. To ensure against these false positives, we filtered genes by Pearson correlation between sample means in PP and LL. Genes with a correlation r>0.85 were not counted as heterochronic.

Heterochronies are a change in the gene expression profile depending on developmental morph, but how different are the patterns of heterochronic changes? Parsimoniously, we would expect most heterochronies to have a similar overall shape that is simply shifted by a stage or two (like a switch from cluster 1 to cluster 2; *Figure 4*). Examining the six clusters above, it is obvious that some expression profiles have similar trajectories (clusters 5 and 6, for example, both show increasing expression post blastula), while others are essentially inverted expression patterns (clusters 1 and 4 for example). Clusters are ordered in a correlation heatmap (*Figure 5A*) from early gene expression to late gene expression. We find that most genes that switch expression profiles between PP and LL do so between the most similar clusters, and thus are near the diagonal. But where we see opposite profiles, the trend is early expression in PP genes compared to LL (top left corner). This is expected given the delay in embryogenesis for LL compared to PP. That said, many genes are expressed earlier in LL compared to PP (bottom right orange triangle in *Figure 5A*) but most of these are relatively small shifts.

We quantify the magnitude of change in gene expression between PP and LL independently of the cluster assignment by estimating the Pearson correlation directly. This shows that most heterochronies are minor changes in the timing of gene expression. There is a left-tailed distribution indicating that a few genes have a large difference in expression patterns (like a profile inversion) between PP and LL (*Figure 5B*).

Gene ontology (GO) enrichment tests show heterochronic genes that are expressed earlier in PP are functionally enriched for mesoderm specification, cell fate specification pathways, and several organogenesis pathways (others include BMP signaling, mesoderm specification, intestinal epithelial cell differentiation, and gene silencing (see *Supplementary file 1*)). Notably, heterochronic genes which are shifted earlier in LL are enriched for many metabolic functions. The shift in expression of metabolic genes is consistent with the life-history and feeding differences between the larvae, and it

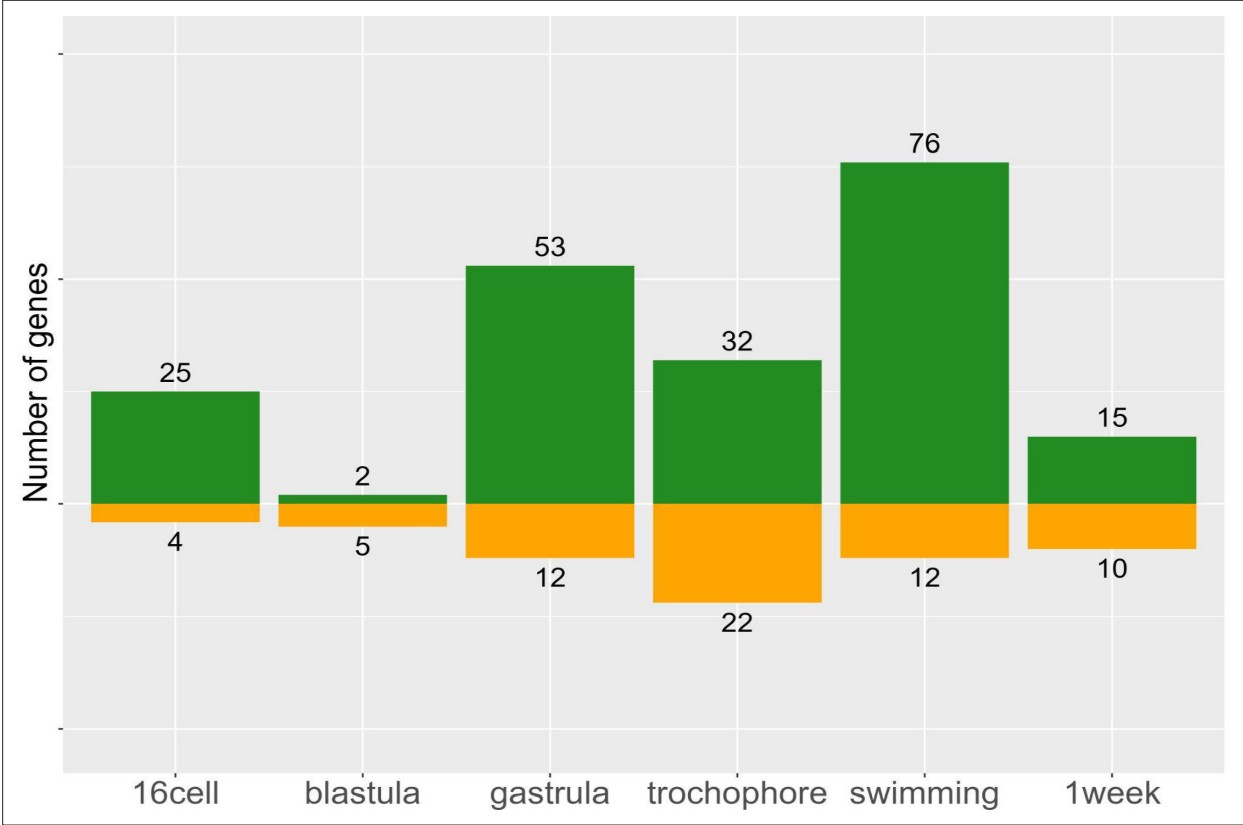

**Figure 6.** Morph-specific gene expression by stage. These genes are never expressed in the opposite morph, but may be expressed at one to six stages within a morph. Planktotrophic (PP) is green and lecithotrophic (LL) is orange.

may be that most of what is differentiating a PP and LL larvae is the onset of feeding and gut-related developmental programs.

Morph-specific genes, by definition, have no expression in one morph and cannot be assigned to a cluster. To quantify morph-specific genes we use a data-adaptive flag method (DAFS) to calculate expression thresholds for each sample, below which genes are not considered as expressed. Using this approach, we identify 195 genes (25.3% of DE genes) which are only expressed in one morph (*Figure 6*). We find considerably more P-specific genes (150 genes in PP vs 45 genes in LL), which tend to be expressed in later developmental stages. These include genes which code for proteins such as Fibropellin, Forkhead-box, Crumbs-like, Notch-2, and many zinc-finger proteins. While the function of these genes during development remains to be determined, it is possible that they maintain PP-specific larval traits.

Expression of functionally distinct categories of genes may evolve by different mechanisms. Morph-specific genes may be involved in very different functional processes than heterochronic genes in this system. GO enrichment tests for morph-specific genes expressed in PP found a few functions related to cell fate specification and signaling pathways (GO enrichment tests for biological process, p<0.05; no significant functional enrichments for LL-specific genes), but a distinct lack of genes involved in metabolic processes. This is because metabolic pathways genes shifted earlier in the development of LL offspring are still required in both offspring morphs. But the morph-specific genes are not required for development and may be modified by different evolutionary mechanisms. These findings also imply that PP embryos require a few specific cell types to produce a PP larval form that are reduced or lost in LL.

Overall, we find that heteromorphic changes in expression accounts for most gene expression differences, which is expected as this is the smallest change in gene expression of the three categories. Gene expression need only be significantly different at one stage to fit this category, and such differences may not necessarily translate into biological differences in development. Morph-specific and heterochronic changes make up very similar proportions of the remaining differences (*Figure 7*).

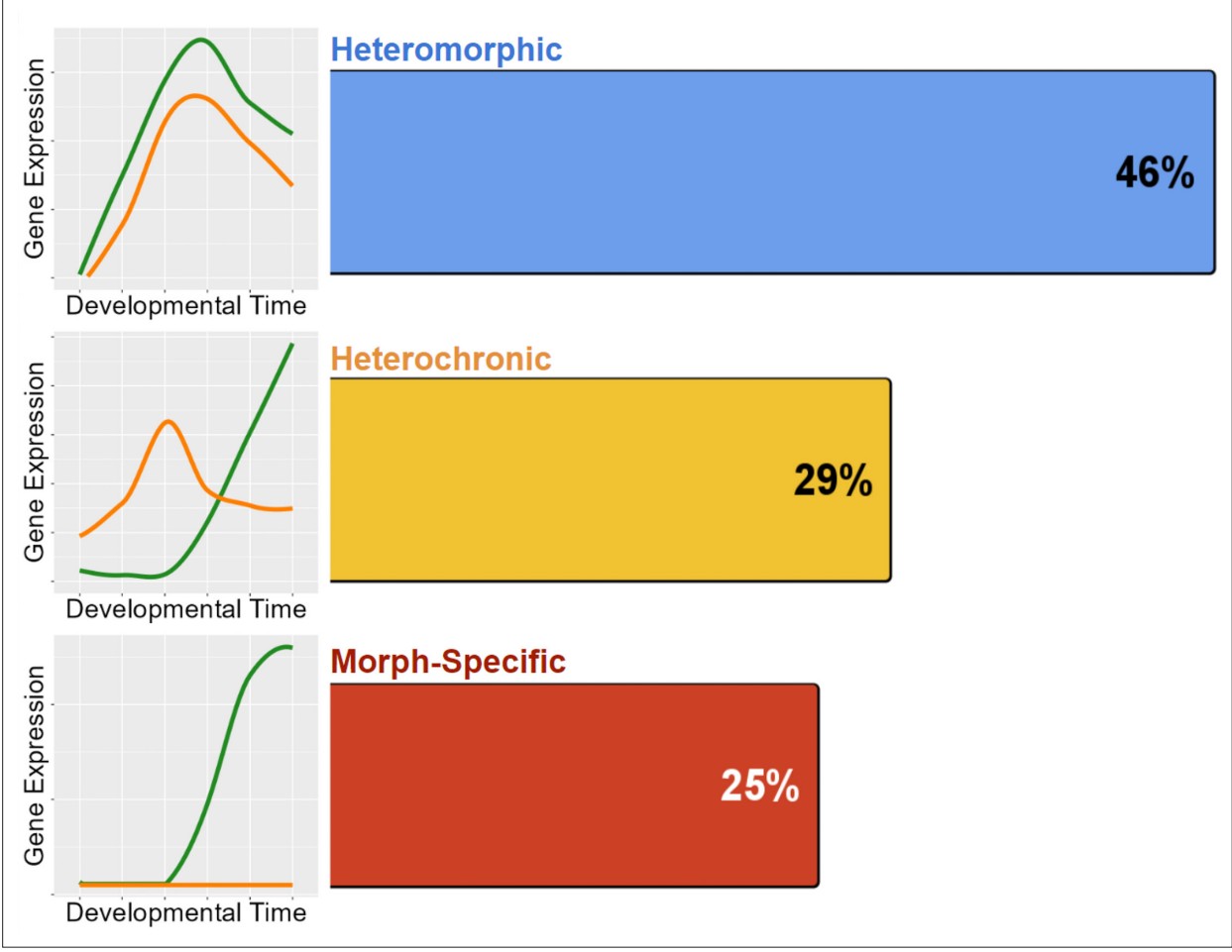

**Figure 7.** Distribution of the 772 genes with differentiated expression between morphs. Heteromorphy makes up nearly half of all differentiated genes, while heterochrony and morph-specific genes split the remainder nearly equally.

This demonstrates that while true heterochronies are common, they are not the main driver of gene expression differences in development.

## Gene expression of genetic crosses

Because we are using an intraspecific comparison, we can extend our analysis to understand the regulatory architecture behind these expression differences. We included RNAseq time course data for offspring from reciprocal crosses between the two morphs, meaning PP and LL parents were crossed in both directions alternating the role of the mother. Cross of the two developmental morphs to produce $F_1$ offspring, which can have a range of intermediate traits, but typically closely resemble their mother's phenotype (*Zakas and Rockman, 2014*). This is particularly useful to disentangle maternal effects, as both $F_1$s (PL and LP; mother's genotype is listed first) are heterozygotes, but they originate from different egg sizes and mothers with different genetic backgrounds. $F_1$s allow us to identify the regulatory architecture underlying DE, and to assess the impact of maternal background on gene expression.

$F_1$s have a general pattern of intermediate expression values but have more variability within replicates and clear cases of outliers (*Figure 8A*, *Figure 8—figure supplement 1*). They have considerably more variability across replicates, and some genes are *misexpressed* – where the $F_1$ expression value is outside the range of the parental difference (*Figure 8A*). Misexpression is reported in hybridization studies and is thought to be the result of epistatic interactions between divergent genomes that reduce the fitness of offspring contributing to sympatric speciation (*Norrström et al., 2011*; *Moran et al., 2021*). Whether misexpression affects the fitness of the embryos in *S. benedicti* is unclear,

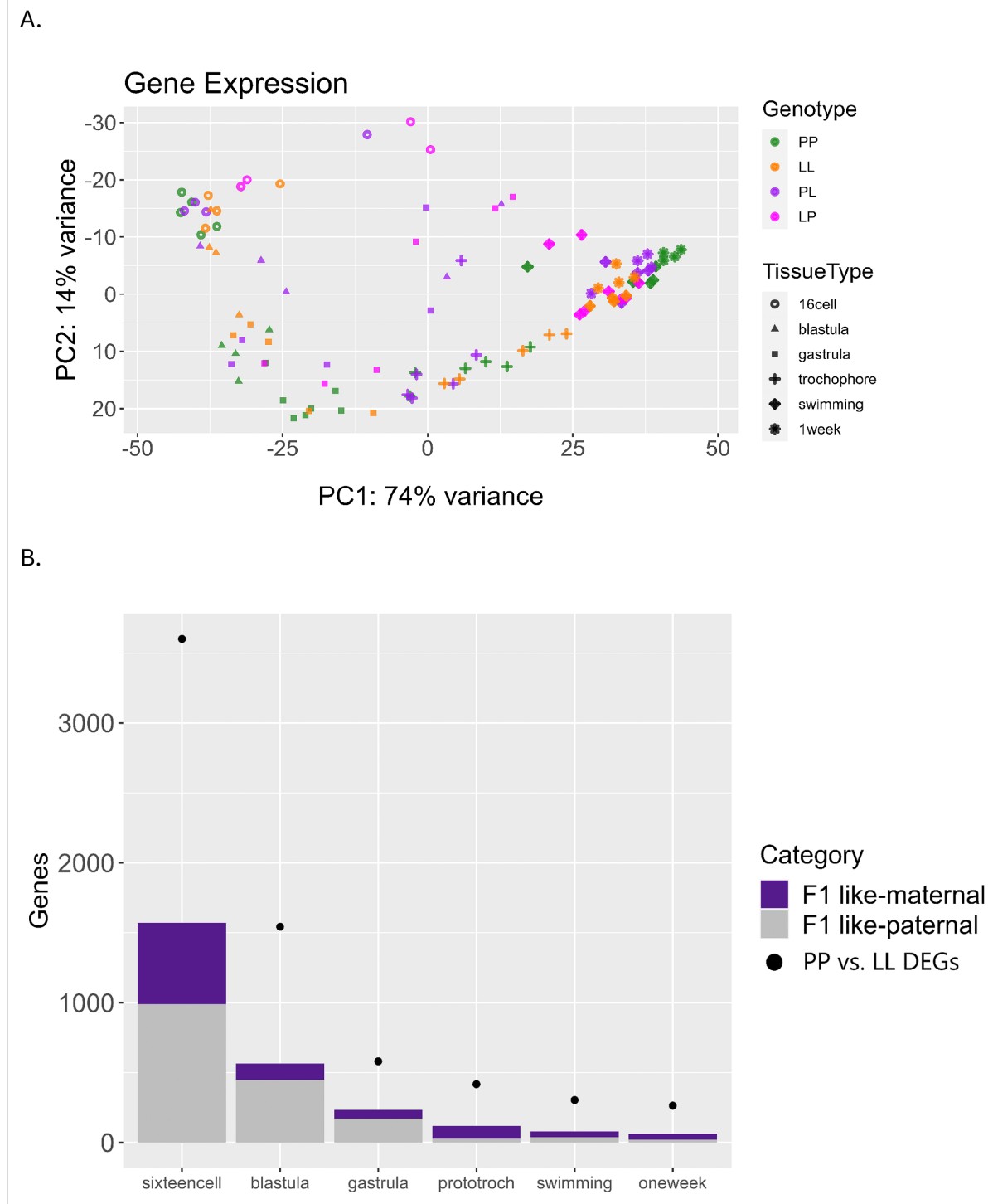

**Figure 8.** Differential gene expression including F1 offspring. (**A**) Principal component analysis (PCA) of the top 500 most variable genes, including $F_1$ (PL or LP) offspring. $F_1$s have more misexpression in early stages but converge after the swimming stage and become intermediate to planktotrophic (PP) and lecithotrophic (LL) samples. (**B**) Plot relating $F_1$ expression back to parental types. Dots indicate the number of DEGs between PP and LL samples at each stage. Purple bars represent the number of those DEGs for which the $F_1$ gene expression is more like its maternal parent, and gray bars represent the number of those DEGs for which the $F_1$ gene expression is more similar to its paternal parent.

The online version of this article includes the following figure supplement(s) for figure 8:

**Figure supplement 1.** Principal component analysis (PCA) of all gene expression labeled by family (crossID).

**Figure supplement 2.** Genes with parental effects.

although F₁s typically develop normally in the lab and no systematic reduction in viability has been observed. In previous studies of gene expression in eggs of the two morphs, we saw a similar pattern where F₁ eggs typically have intermediate expression values compared to parental genotypes, but high levels of misexpression (*Harry and Zakas, 2023*). For genes that are DE between PP and LL samples, we find that F₁ gene expression patterns are split between those matching the maternal and paternal gene expression pattern, with slightly more genes matching the paternal expression at each stage (*Figure 8B*). We expect most transcripts in the first and second stages of development to be maternally derived, so the high proportion of genes matching paternal expression patterns is somewhat unexpected.

We find that most genes (>8000 genes on average, out of 14,327 genes expressed) have extremely conserved expression patterns among PP, LL, and F₁ embryos at each developmental stage. The high variability and misexpression of F₁s negatively impacts our statistical power by introducing variation and results in fewer genes being confidently assigned gene regulatory mechanisms. As a result, almost no genes can be identified as having a dominant or additive inheritance pattern, while 150 genes are identified as overdominant over the course of development. Notably, very few morph-specific genes are expressed at a significant level in either F₁s (between 0 [PP] and 3 [LL] genes), strongly suggesting negative regulation of expression that acts in trans for these genes.

## Regulatory architecture

We leverage the F₁ offspring to dissect the regulatory architecture underlying developmental gene expression differences; we use allele-specific expression patterns in F₁ offspring – tracking the expression of the maternal or paternal allele – to assign gene regulatory differences as either *cis*- or *trans*-acting modifications (or both; *Davidson and Peter, 2015*). Typically this approach is used in hybrids (*Wittkopp et al., 2004*; *Tirosh et al., 2009*; *McManus et al., 2010*; *Coolon et al., 2014*; *Wang et al., 2020*), but we have adapted it to intraspecific F₁ offspring to test whether early divergence is consistent with predictions in the literature (*Harry and Zakas, 2023*).

We determine the regulatory architecture underlying DE by calculating P and L allele-specific expression in reads from F₁ samples and assigned primary regulatory modes according to established

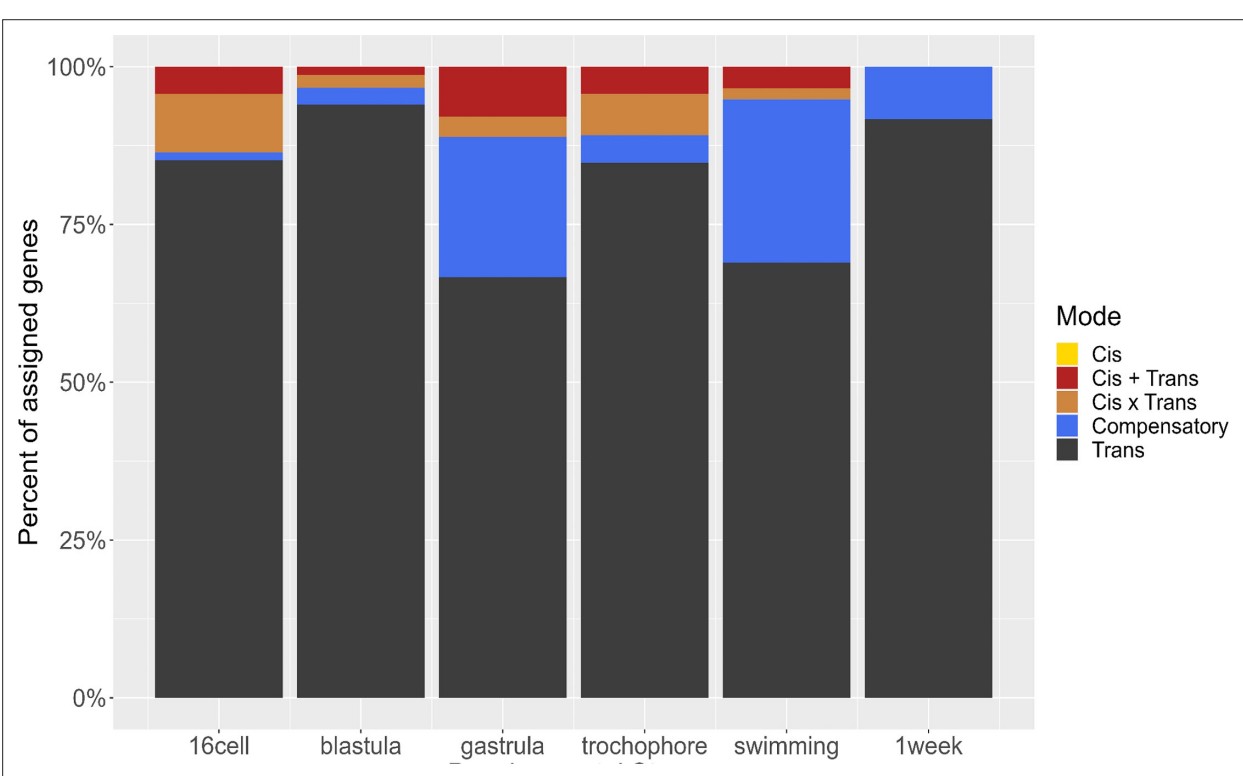

**Figure 9.** Distribution of regulatory mode over time. The majority of regulatory changes act in trans, and compensatory changes emerge after gastrulation.

criteria (*Wittkopp et al., 2004*; *Graze et al., 2009*; *Coolon et al., 2014*; *Wang et al., 2020*). As this is an intraspecific comparison with little fixed genomic differences between the morphs, not all genes have alleles that can be assigned parentage; however, a set of ~2000 DE genes have morph-differentiating SNPs enabling the parental assignment of $F_1$ transcript reads. The regulatory modes of these DE genes are assigned as '*cis*', '*trans*', 'cis+trans', 'cis × trans', or 'compensatory' (see *Supplementary file 2* for formal criteria). Of this set, misexpression in $F_1$ offspring causes many genes to receive ambiguous assignments. As a result, only 143 genes have informative regulatory mode assignments. While this is a small number, we expect that the genes we can classify are proportionally representative of the remaining regulatory architecture.

The primary mode of regulatory change throughout development is *trans*-acting (*Figure 9*). No differences are caused by purely *cis*-acting regulatory modifications, though we did find some cis-×-trans and cis-×-trans interactions. We found that the number of genes with compensatory regulatory modifications increases sharply after gastrulation, indicating that gene expression may be more tightly controlled past that point. Studies across *Drosophila* species, for example, have indicated that maternal effects are regulated differently than zygotic effects, but in this case there are larger *trans*-acting factors in early (maternally controlled) developmental stages (*Cartwright and Lott, 2020*). We do not see such a clear effect; *trans*-acting factors make up most of the regulatory architecture throughout development. Interestingly, this result is different from previous studies of egg mRNA expression of this species, where *cis*- and *trans*-regulatory modifications were found at similar rates for maternal gene expression (*Harry and Zakas, 2023*). Comparison across the egg (maternal) and embryo (maternal and zygotic) regulatory landscape shows greater *cis*-acting regulation in the maternally expressed genes. These results demonstrate the possibility that maternal and zygotic gene regulatory architecture evolve through distinct mechanisms and on separate timescales.

We tested the possibility that the reciprocal $F_1$s generated in the study (for the three stages that have both LP and PL samples) might have different regulatory mode changes due to parental effects. Splitting our analyses into separate PL and LP parental arrangements did not yield significantly different results, but treating them separately meant fewer genes could be assigned a mode.

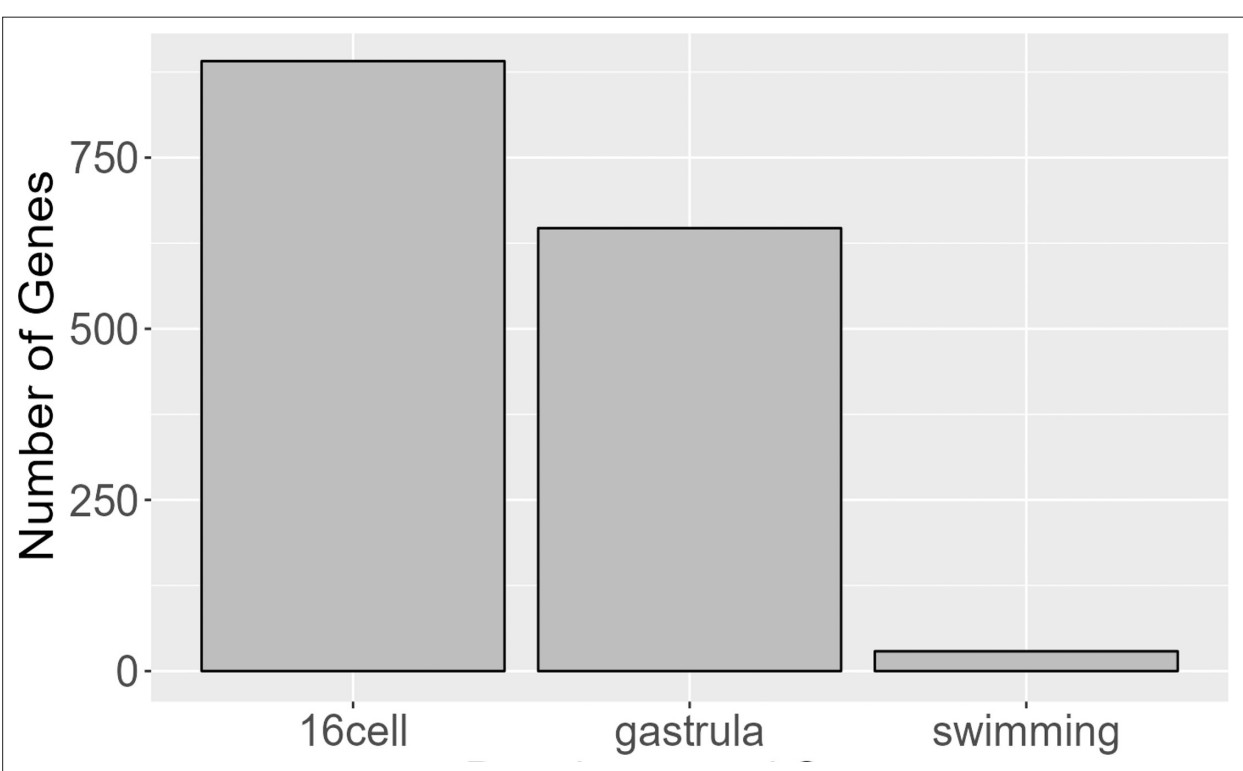

**Figure 10.** Genes with parental effects. Early developmental stages have significantly more genes, likely due to maternal transcript differences. Maternal transcripts are likely degraded around the gastrula stage and any remaining parental effects observed are likely due to other mechanisms.

We also investigated parent-of-origin effects on gene expression using $F_1$s. These occur when an allele's expression is at least partially dependent on which parent contributed that allele. For instance, maternal effects may involve polymorphisms that affect the development of an offspring only when contributed as part of the maternal genome (and which impose no developmental variation when contributed by the paternal genome). These effects are a form of epigenetic inheritance which can allow for complex, adaptive maternal-zygote interactions such as certain metabolic genes being specifically activated in the offspring to match the nutrient content of its mother's egg. Due to animal constraints where LL mothers produce far fewer offspring per clutch (10–40 on average), we were only able to collect samples for $F_1$ offspring with L mothers (LP) for three developmental stages (16-cell, gastrula, and swimming). All six time points were sampled for $F_1$s with PP mothers (PL). To find parental effects (maternal or paternal), we conducted a differential expression analysis contrasting the PL samples to the LP samples. Tests for differential expression show that many significant differences in gene expression between the $F_1$ morphs arise at early stages (ostensibly due to the presence of maternal transcripts inherited through the egg) and then drop sharply after gastrulation (*Figure 10*; *Figure 8—figure supplement 2*). This indicates that while a few parental effects on gene expression may persist throughout development, most parental effects are limited to early embryogenesis, which is consistent with expectations for maternal transcription degradation.

## Discussion

By comparing gene expression at the relative embryological stages of the two morphs, we determine the extent that expression changes contribute to life-history differences. LL offspring take longer to reach the same embryonic developmental stages as PP in absolute time, although the stages are similar, and morphological differences are not observed until later development. Numerous morphological and life-history differences occur between the morphs at the swimming larval stage (reviewed in *Zakas, 2022*), but here we detail differences in the embryology as well. Evolutionarily, we expect that LL is the more derived developmental mode, which likely arose from a maternal increase in egg size followed by adaptive changes in the genome that increased LL fitness (reviewed in *Zakas, 2022*). This generates predictions about the types of genes that we expect to change in patterns and magnitude. For example, we may expect that genes associated with feeding processes, growth, and cell cycle and specification initiate faster in the smaller, PP embryo (*Wray and Lowe, 2000*; *Strathmann, 2000*; *Lowe et al., 2002*; *Segers et al., 2012*; *Figueroa et al., 2021*). The non-obligatory-feeding LL larvae have a delay in mesoderm formation and through-gut development (*Pernet, 2003*; *Gibson et al., 2010*) so we would expect a delay in LL for mesoderm and gut specification, which we do see in the GO terms for the genes expressed earlier in PP.

The pattern of differential expression we observe over development – where numerous but small expression differences occur early followed by a few genes with large expression differences (*Figure 2C*) – is consistent with the importance of gastrulation to canalize development. Gastrulation is the 'phylotypic stage' at which different phyla are the most morphologically and molecularly similar to each other during their development (*Slack et al., 1993*; *Duboule, 1994*; *Richardson, 1995*; *Irie and Kuratani, 2011*; *Irie and Kuratani, 2014*; *Macchietto et al., 2017*). Furthermore, adults of both types are extremely similar, so we would not expect persistent, large-scale gene expression differences at these late stages. The few genes that remain significantly DE post gastrulation are candidates for maintaining morphological and life-history differences between the morphs. (Chitin biosynthesis genes, for example, remain significantly different at later stages, and upregulated in PP.) However, these genes' expression levels might converge much later in development as both morphs reach adulthood.

We see similar amounts of heterochronic and morph-specific genes. Heterochronic genes, and their pathways, are ideal candidates for modifying larval and life-history traits. Morph-specific genes are of particular interest in *S. benedicti* as previous genetic and transcriptomic work suggests there is very little differentiation of the two morphs at the genomic level (*Zakas et al., 2018*; *Zakas et al., 2022*). While it is possible that genomic rearrangements (*Janssen et al., 2001*), gene duplications (*Long et al., 2003*), or co-options (*Linksvayer and Wade, 2005*) underlie the expression of morph-specific genes, given previous findings of limited total sequence differentiation between morphs (*Zakas et al., 2018*; *Zakas et al., 2022*), we find these possibilities unlikely. The morph-specific genes

are more likely driven by regulatory differences in enhancing or silencing elements (as in *Zhao et al., 2017*; *Matlosz et al., 2022*).

We expect that incipient species that are early in their divergence have more genetic regulatory changes due to *trans*-acting factors; this is because *trans*-acting factors may be more pleiotropic and initiate numerous developmental changes parsimoniously. Essentially a few *trans*-acting regulatory modifications, such as changes to transcription factors, could account for most of the developmental and phenotypic differences between morphs. As species divergence time increases, *cis*-acting elements can arise and refine gene expression of individual genes (*Wittkopp et al., 2004*; *Coolon et al., 2014*; *Cartwright and Lott, 2020*). These regulatory architecture trends have been reported in other species, such as the urchin *Heliocidaris* spp. where a PP and LL species have diverged in the same genus (*Israel et al., 2016*; *Wang et al., 2020*). However, no studies to date have examined this regulatory architecture in the context of the short evolutionary window of divergent morphs within the same species. This suggests the life-history differences we see in this early evolutionary divergence are driven by a few developmentally upstream trans-acting factors with pleiotropic effects.

We quantify the relative contribution of heteromorphic, heterochronic, and morph-specific gene expression pattern modifications in the evolution of a developmental dimorphism. We find that heterochronic and morph-specific genes contribute similarly to gene expression differentiation. Additionally, we find that the regulatory architecture of differential expression is predominantly *trans*-acting modifications, supporting the hypothesis that early gene expression evolution occurs by few, highly pleiotropic *trans*-acting regulatory modifications.

## Methods
### Animal rearing and sample collection
We use lab-reared male and female *S. benedicti* originally sampled from Newark Bay Bayonne, New Jersey (PP), and Long Beach, California (LL). All experimental procedures and growth incubations are carried out at 20°C unless otherwise noted. To produce offspring for sampling, we crossed virgin females with one male for each test cross (Cross ID in *Figure 2A*). PP offspring were fed small quantities of our lab's standard feeding algae after the swimming stage had been reached to avoid starvation. As clutch sizes can be quite small, we took advantage of the multiple broods that can be produced by a single mating event. We used three to five consecutive broods per female where all offspring were full sibs. This is necessary as embryo number is limited (10–40 embryos per clutch for LL and 100–400 for PP) and 10–40 pooled embryos are required to produce sufficient input mRNA. In this study brood number and developmental stage are confounded within sample groups (Cross IDs).

### Embryo timeline construction
To build a timeline of development, we removed embryos from the maternal brood pouch and observed development in a Petri dish in artificial seawater in an incubator at 20°C (previous work indicates development is normal outside the brood pouch). We selected six distinct time points that we confirmed with cell counts (nuclei) using Hoechst: 16 cells, blastula (64 cells), and gastrula (124 cells). Time to each development stage was averaged over at least four clutch observations. Later embryo and larval stages were identified by morphological differences (appearance of ciliated band trochophores and eye development). Six stages capture a broad scope of developmentally critical periods while having enough timing separation to be distinct (each embryonic time point was ~12 hr apart). Images were captured using a ×40 objective on a Zeiss Axio inverted microscope with an indicator for scale which was subsequently used to scale images to the same relative size in *Figure 1*.

### RNAseq
Embryos were collected from a single female, and 10–40 were processed for total mRNA extraction at each time point. If the number of embryos was insufficient for all time points, we used multiple broods. We used a minimum of 20 LL embryos for stages 16-cell, blastula, gastrula, and 10 LL embryos for stages trochophore, swimming, 1 week. PP females have large clutches, so embryos were divided equally (~40 embryos/clutch) among the six stages. With this approach we could not collect all time points for the same females, but there are at least two complete sets and at least four replicates per

stage for each morph. The same sampling stages were used when collecting $F_1$ samples in this experiment (**Figure 2A**).

Embryo RNA extraction used the Arcturus PicoPure kit including the DNAse step. We used a Qubit RNA kit to measure RNA yields for pilot data, but once established we bypassed this step to maximize RNA yields. Libraries were constructed with the NEB UltraII Stranded RNA library prep kit (cat# E7760S) for Illumina. Libraries were sequenced on two lanes of 150 bp on the Illumina NovaSeq.

## Sequencing read quality trimming and mapping

We used TrimGalore (cutadapt) (**Martin, 2011**) and FastP (**Chen et al., 2018**) for quality assessment and trimming. Reads were mapped to a reference of all transcript sequences (transcripts extracted with GFFread, **Pertea and Pertea, 2020**) that are annotated in the *S. benedicti* reference genome (**Zakas et al., 2022**) using Salmon 1.10 (**Patro et al., 2017**) using the default scoring parameters. The reference genome for *S. benedicti* is a chromosome-level assembly with >99% of genes occurring in the first 11 chromosomal scaffolds (**Zakas et al., 2022**). Individual sample transcript expression quantification estimates were summarized to the gene-level using the Tximport R package (**Soneson et al., 2015**).

## Library diagnostics

As the *S. benedicti* genome is from P individuals (**Zakas et al., 2022**), we checked the mapping rate of P and L samples (as well as both $F_1$s) at each developmental stage to ensure similar mapping rates (**Figure 2—figure supplement 1**). We calculated mapping by dividing the sum of expression estimates for all genes by the sequencing depth for each sample individually, and then averaging samples from each developmental stage by morph to produce an average mapping rate (**Figure 2—figure supplement 1**). We find LL and PP mapping rates do not differ significantly (two-sided t-test; p=0.1118), nor do the $F_1$ samples' mapping rates differ from any other group (**Figure 2—figure supplement 2**). While this does not eliminate the possibility of mapping bias, these results indicate that missing data would not substantially change our results.

Samples which received fewer than 2 million total reads were excluded from further analyses since our expected sequencing depth was between 20 and 40 million reads (six samples total). Samples were normalized using the DESeq2 R package (**Love et al., 2014**) and expression estimates were transformed using the variance stabilizing transformation function to perform PCA using the function *prcomp* (**R Development Core Team, 2020**), which is plotted for the first two principal components with ggplot2 (**Wickham, 2016**; **Figure 2B**).

## Differential expression

We used DESeq2 to determine significance in expression differences (**Love et al., 2014**) using the variable grouping method recommended by the DESeq2 manual. We used two factors for characterization: developmental stage and genotype (P, L, PL, LP) which generate a factor level for each stage-Head2/genotype combination (i.e. '16-cell_Lecithotroph') which we called 'multiFactor' for DESeq2 as 'design = ~multiFactor'. The Benjamini-Hochberg false discovery rate (FDR) algorithm was used with p-values between factor levels of 'multiFactor' to reduce the incidence of false positives for differential expression. Throughout this study, the threshold for significant gene expression differences is an FDR-adjusted p-value of 0.05 or less, and an expression fold-change greater than twofold.

## Expression profile clustering

To summarize expression clusters we used the Mfuzz 2.58 R package (**Futschik and Carlisle, 2005**; **Kumar and E Futschik, 2007**). Clusters are based on mean expression estimates from P samples. Normalization was (DESeq2, **Love et al., 2014**), log2-transformed, and then filtered for genes with low variability/expression and standardized using the Mfuzz functions '*filter.std*' and '*standardize*', respectively. Mfuzz requires a priori cluster numbers. To capture the representative sample of expression patterns, we found the highest number of clusters for which Pearson correlations of pairs of cluster centroids (the single most cluster-representative gene for each cluster) did not exceed 0.85 (r<0.85) for any pair of clusters. This resulted in six clusters. The fuzzifier coefficient (m) required by Mfuzz was estimated to be 1.71 by the function '*mestimate*' (**Futschik and Carlisle, 2005**; **Kumar and E Futschik, 2007**). Following cluster generation with the P expression data, the L expression data was

**Table 1.** Logical table showing criterion for morph-specific expression classification of genes based on number of samples in each group that have gene expression greater than their expression level threshold as assessed by data-adaptive flag method (DAFS).

An allowance for a single mismatched replicate is made in this criterion. This is because we do not believe that a single incidence within a morph (across all developmental stages) is a functionally significant and biologically relevant level of gene expression in this case.

| PP samples above expression threshold | LL samples above expression threshold | Classification |
|---|---|---|
| ≤1 | ≥3 | Lecithotrophic-specific expression |
| ≥3 | ≤1 | Planktotrophic-specific expression |

mapped onto the clusters using the Mfuzz function '*membership*' so both morphs could be compared. Clusters were plotted using the function '*mfuzz.plot2*' (*Figure 4*, *Figure 4—figure supplement 2*).

## Expression classification

To identify genes as heterochronic we first identify genes that have different expression patterns between the morphs and then filter those genes based on their Pearson correlation coefficients. Genes for which the PP and LL expression are on different clusters are categorized as heterochronic. Some of these genes fall onto different clusters but still have relatively similar expression profiles. Therefore, we also filter by a measure of the Pearson correlation coefficient. The threshold is based on the distribution of correlation coefficients of all PP to LL genes' expression in the dataset (*Figure 5B*). Based on the distribution, a threshold of r<0.85 was selected to filter genes which would be classified as switching clusters but are similar enough in their expression between the two morphs to be removed. This filter disqualified 38 genes.

## Morph-specific expression

To categorize morph-specific expression we use a gene expression thresholding approach. Spurious gene expression resulting in low levels of estimated expression (few RNAseq reads) is categorically and functionally distinct from robust gene expression (*Hebenstreit et al., 2011*). We establish meaningful expression thresholds based on the assumption that genes with robust expression will have values which are normally distributed. Individual sample thresholds were calculated using a DAFS (*George and Chang, 2014*). Thresholds were used to identify genes which are expressed at significant levels in only PP or only LL offspring. For example, if a gene is expressed above its threshold in three or more (out of five) P samples at one or more developmental stages and in no more than one LL sample at any developmental stage, then that gene was categorized as morph-specific (*Table 1*).

## F$_1$ expression time-series collection

We concurrently sequenced RNA libraries from F$_1$ offspring (*Figure 2A*). We generated F$_1$ offspring from reciprocal crosses P×L and L×P (both mother-father directions) using the same sample collection method as stated above. However, we only sequenced samples from three of the developmental stages for the F$_1$s in the LP direction. All analyses were performed as above.

## Mode of inheritance and F$_1$ misexpression

Using F$_1$ data, we classified the mode of inheritance for each gene according to established criteria and differential expression tests from DESeq2 (*Coolon et al., 2014*; *Harry and Zakas, 2023*; *Wang et al., 2020*). Genes which were significantly DE between reciprocal F$_1$ offspring (PL vs LP) at any developmental stage were not considered for this analysis. The remaining genes were classified as either (1) conserved, (2) additive, (3) dominant for one genotype, or (4) misexpressed (over/under-dominant).

## Parent of origin effects

DE genes between PL and LP offspring are different due to parental effects. DE was detected with DESeq2 with the contrasts 'c("multiFactor", "PL_sixteencell", "LP_sixteencell")', 'c("multiFactor", "PL_gastrula", "LP_gastrula")', and 'c("multiFactor", "PL_swimming", "LP_swimming")' similar to PP

and LL samples. We evaluate these at three of the six developmental stages. The direction of the parental effect was determined by matching the expression change with the direction of each identified gene's expression in PP and LL samples (where those genes were DE in PP and LL samples).

### Mode of regulatory change

To measure allele-specific expression, we assigned sequencing reads from $F_1$ samples to either P or L parentage by identifying fixed SNPs within the transcript sequences of the parental types. We used HyLiTE (*Duchemin et al., 2015*) to identify SNPs and assign reads as a P or L allele. We then categorized genes' regulatory mode according to established empirical methods (*Wittkopp et al., 2004*; *Graze et al., 2009*; *Coolon et al., 2014*; *Harry and Zakas, 2023*; *Wang et al., 2020*) for each developmental stage independently. This required three comparisons of each gene's expression which we performed using DESeq2: (1) the contrast of PP to LL samples, (2) the contrast of P alleles to L alleles within $F_1$ samples, and (3) a ratio of the differential expression of PP:LL to the differential expression of P:L alleles. For (3) we use a special contrast in DESeq2, applying the design (~Geno × Ori) where Geno identifies reads as either a P or L allele and Ori identifies the reads as originating from the parentals or $F_1$ samples. DE genes were categorized as either in 'cis', 'trans', 'cis+trans', or 'cis × trans' (*Wittkopp et al., 2004*; *Graze et al., 2009*; *Coolon et al., 2014*; *Harry and Zakas, 2023*; *Wang et al., 2020*).

## Acknowledgements

Sarah Cole for help with collecting embryos, animal rearing, constructing the sampling timeline. Thanks to Matthew Rockman and Greg Wray and members of the Zakas lab for comments on the manuscript. This work was supported by National Institutes of Health NIGMS grant 5R35GM142853 to C Zakas.

## Additional information

### Funding

| Funder | Grant reference number | Author |
| --- | --- | --- |
| National Institutes of Health | 5R35GM142853 | Christina Zakas |

The funders had no role in study design, data collection and interpretation, or the decision to submit the work for publication.

### Author contributions

Nathan D Harry, Resources, Data curation, Formal analysis, Investigation, Visualization, Methodology, Writing – original draft, Writing – review and editing; Christina Zakas, Conceptualization, Funding acquisition, Investigation, Visualization, Methodology, Writing – original draft, Project administration, Writing – review and editing

### Author ORCIDs

Nathan D Harry http://orcid.org/0000-0001-9271-6636
Christina Zakas https://orcid.org/0000-0002-9998-848X

Reviewer #1 (Public Review): https://doi.org/10.7554/eLife.93062.3.sa1
Reviewer #2 (Public Review): https://doi.org/10.7554/eLife.93062.3.sa2
Author response https://doi.org/10.7554/eLife.93062.3.sa3

## Additional files

### Supplementary files

• Supplementary file 1. Gene ontology (GO) enrichment results spreadsheet.
• Supplementary file 2. Mode of regulatory change classifications.

• MDAR checklist

## Data availability

All reads are submitted to NCBI under BioProject: PRJNA1008044. All analyses and datasets are available at https://github.com/NathanDHarry/Harry-Zakas-TC, (copy archived at *Harry, 2024*) a GitHub repository which contains estimated transcript expression data generated by Salmon 1.10 and an R script file containing annotated code used to conduct these analyses.

The following dataset was generated:

| Author(s) | Year | Dataset title | Dataset URL | Database and Identifier |
|---|---|---|---|---|
| Harry ND, Zakas C | 2024 | Streblospio benedicti RNAseq developmental time-series | https://www.ncbi.nlm.nih.gov/bioproject/PRJNA1008044 | NCBI BioProject, PRJNA1008044 |

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
