## [Editor Report · eLife assessment]

This **important** study examines the extent to which distinct developmental pathways that result in alternative morphs correlate with transcriptome differences in a marine annelid, Streblospio benedicti. The strengths of the study include the experimental design and dense temporal sampling, which together provide **convincing** evidence that the two morphs can be clearly distinguished at the transcriptome level, despite relatively modest overall differences. The work will be of particular interest to students of the evolution of development.

---

## [Referee Report · Reviewer #1 (Public Review)]

Summary:

Overall, this study provides a meticulous comparison of developmental transcriptomes between two sub-species of the annelid Streblospio benedicti. Different lineages of S. benedicti maintain one of two genetically programmed alternative life histories, the ancestral planktotrophic or derived lecithotrophic forms of development. This contrast is also seen at the inter-species level in many marine invertebrate taxa, such as echinoderms and molluscs. The authors report relatively (surprisingly?) modest differences in transcriptomes overall, but also find some genes whose expression is essentially morph-specific (which they term "exclusive").

Strengths:

The study is based on dense and appropriately replicated sampling of early development. The tight clustering of each stage/morph combination in PCA space suggests the specimens were accurately categorized. The similar overall trajectories of the two morphs was surprising to me for two stage: (1) the earliest stage (16-cell), at which we might expect maternal differences due to the several-fold difference in zygote size, and (2) the latest stage (1-week), where there appears to be the most obvious morphological difference. This is why we need to do experiments!

The examination of F1 hybrids was another major strength of the study. It also produced one of the most surprising results: though intermediate in phenotype, F1 embryos have the most distinct transcriptomes, and reveal a range of fixed, compensatory differences in the parental lines. Further, the F1 lack expression of nearly all transcripts identified as morph-specific in the pure parental lines. Since the F1 larvae present intermediate traits combining the features of both morphs, this implies that morph-specific transcripts are not actually necessary for morph-specific traits. This is interesting and somewhat counter to what one might naively expect.

Weaknesses:

Overall I really enjoyed this paper, and in its revised form it addresses some concerns I had in the first version. I still see a few places where it can be tightened and made more insightful.

---

## [Referee Report · Reviewer #2 (Public Review)]

The manuscript by Harry and Zakas determined the extent to which gene expression differences contribute to developmental divergence by using a model that has two distinct developmental morphs within a single species. Although the authors did collect a valuable dataset and trends in differential expression between the two morphs of S. benedicti were presented, we found limitations about the methods, system, and resources that the authors should address.

We have two major points:

(1) Background information about the biological system needs to be clarified in the introduction of this manuscript. The authors stated that F1 offspring can have intermediate larval traits compared to the parents (Line 81). However, the authors collected F1 offspring at the same time as the mother in the cross. If offspring have intermediate larval traits, their developmental timeline might be different than both parents and necessitate the collection of offspring at different times to obtain the same stages as the parents. Could the authors (1) explain why they collected offspring at the same time as parents given that other literature and Line 81 state these F1 offspring develop at intermediate rates, and (2) add the F1 offspring to Figure 1 to show morphological and timeline differences in development?

Additionally, the authors state (Lines 83-85) that they detail the full-time course of embryogenesis for both the parents and the F1 crosses. However, we do not see where the authors have reported the full-time course for embryogenesis of the F1 offspring. Providing this information would shape the remaining results of the manuscript.

(2) We have several concerns about the S. benedicti genome and steps regarding the read mapping for RNA-seq:

The S. benedicti genome used (Zakas et al. 2022) was generated using the PP morph. The largest scaffolds of this assembly correspond to linkage groups, showing the quality of this genome. The authors should point out in the Methods and/or Results sections that the quality of this genome means that PP-specific gene expression can be quantified well. However, the challenges and limitations of mapping LL-specific expression data to the PP genome should be discussed.

It is possible that the authors did not find exclusive gene expression in the LL morph because they require at least one gene to be turned on in one morph as part of the data-cleaning criteria. Because the authors are comparing all genes to the PP morph, they could be missing true exclusive genes responsible for the biological differences between the two morphs. Did they make the decision to only count genes expressed in one stage of the other morph because the gene models and mapping quality led to too much noise?

The authors state that the mapping rates between the two morphs are comparable (Supplementary Figure 1). However, there is a lot of variation in mapping the LL individuals (~20% to 43%) compared to the PP individuals. What is the level of differentiation within the two morphs in the species (pi and theta)? The statistical tests for this comparison should be added and the associated p-value should be reported. The statistical test used to compare mapping rates between the two morphs may be inappropriate. The authors used Salmon for their RNA alignment and differential expression analysis, but it is possible that a different method would be more appropriate. For example, Salmon has some limitations as compared to Kallisto as others have noted. The chosen statistical test should be explained, as well as how RNA-seq data are processed and interpreted.

What about the read mapping rate and details for the F1 LP and PL individuals? How did the offspring map to the P genome? These details should be included in Supplementary Figure 1. Could the authors also provide information about the number of genes expressed at each stage in the F1 LP and PL samples in S Figure 2? How many genes went into the PCA? Many of these details are necessary to evaluate the F1 RNA-seq analyses.

Generally, the authors need to report the statistics used in data processing more thoroughly. The authors need to report the statistics used to (1) process and evaluate the RNA-seq data and (2) determine the significance between the two morphs (Supplementary Figures 1 and 2).

---

## [Author Response]

The following is the authors’ response to the original reviews.

**Reviewer #1 (Public Review):**
Summary:Overall, this study provides a meticulous comparison of developmental transcriptomes between two sub-species of the annelid Streblospio benedicti. Different lineages of S. benedicti maintain one of two genetically programmed alternative life histories, the ancestral planktotrophic or derived lecithotrophic forms of development. This contrast is also seen at the inter-species level in many marine invertebrate taxa, such as echinoderms and molluscs. The authors report relatively (surprisingly?) modest differences in transcriptomes overall but also find some genes whose expression is essentially morph-specific (which they term "exclusive").Strengths:The study is based on a dense and appropriately replicated sampling of early development. The tight clustering of each stage/morph combination in PCA space suggests the specimens were accurately categorized. The similar overall trajectories of the two morphs were surprising to me for two stages: (1) the earliest stage (16-cell), at which we might expect maternal differences due to the several-fold difference in zygote size, and (2) the latest stage (1-week), where there appears to be the most obvious morphological difference. This is why we need to do experiments!The examination of F1 hybrids was another major strength of the study. It also produced one of the most surprising results: though intermediate in phenotype, F1 embryos have the most distinct transcriptomes, and reveal a range of fixed, compensatory differences in the parental lines.Weaknesses:Overall I really enjoyed this paper, but I see a few places where it can be tightened and made more insightful. These relate to better defining the basis for "exclusive" expression (regulation or gene presence/absence?), providing more examples of how specific genes related to trophic mode behave, and placing the study in the context of similar work in other phyla.

As suggested, we changed the term “exclusive expression” to “morph-specific” expression throughout the paper to clarify which genes are only expressed in one morph. We also added references to similar work in other phyla such as recent work on lecithotrophic and planktotrophic development in species of Heliocidaris sea urchins in the 4th paragraph of the discussion. We added additional data about the F1 hybrids in “Gene expression of Genetic Crosses” section and the new Figure 8B. We find that gene expression in F1 offspring is divided between matching the maternal and paternal gene expression patterns, with slightly more genes matching paternal expression.

**Reviewer #2 (Public Review):**
The manuscript by Harry and Zakas determined the extent to which gene expression differences contribute to developmental divergence by using a model that has two distinct developmental morphs within a single species. Although the authors did collect a valuable dataset and trends in differential expression between the two morphs of S. benedicti were presented, we found limitations about the methods, system, and resources that the authors should address.We have two major points:(1) Background information about the biological system needs to be clarified in the introduction of this manuscript. The authors stated that F1 offspring can have intermediate larval traits compared to the parents (Line 81). However, the authors collected F1 offspring at the same time as the mother in the cross. If offspring have intermediate larval traits, their developmental timeline might be different than both parents and necessitate the collection of offspring at different times to obtain the same stages as the parents. Could the authors (1) explain why they collected offspring at the same time as parents given that other literature and Line 81 state these F1 offspring develop at intermediate rates, and (2) add the F1 offspring to Figure 1 to show morphological and timeline differences in development?Additionally, the authors state (Lines 83-85) that they detail the full-time course of embryogenesis for both the parents and the F1 crosses. However, we do not see where the authors have reported the full-time course for embryogenesis of the F1 offspring. Providing this information would shape the remaining results of the manuscript.(2) We have several concerns about the S. benedicti genome and steps regarding the read mapping for RNA-seq:The S. benedicti genome used (Zakas et al. 2022) was generated using the PP morph. The largest scaffolds of this assembly correspond to linkage groups, showing the quality of this genome. The authors should point out in the Methods and/or Results sections that the quality of this genome means that PP-specific gene expression can be quantified well. However, the challenges and limitations of mapping LL-specific expression data to the PP genome should be discussed.It is possible that the authors did not find exclusive gene expression in the LL morph because they require at least one gene to be turned on in one morph as part of the data-cleaning criteria. Because the authors are comparing all genes to the PP morph, they could be missing true exclusive genes responsible for the biological differences between the two morphs. Did they make the decision to only count genes expressed in one stage of the other morph because the gene models and mapping quality led to too much noise?The authors state that the mapping rates between the two morphs are comparable (Supplementary Figure 1). However, there is a lot of variation in mapping the LL individuals (~20% to 43%) compared to the PP individuals. What is the level of differentiation within the two morphs in the species (pi and theta)? The statistical tests for this comparison should be added and the associated p-value should be reported. The statistical test used to compare mapping rates between the two morphs may be inappropriate. The authors used Salmon for their RNA alignment and differential expression analysis, but it is possible that a different method would be more appropriate. For example, Salmon has some limitations as compared to Kallisto as others have noted. The chosen statistical test should be explained, as well as how RNA-seq data are processed and interpreted.What about the read mapping rate and details for the F1 LP and PL individuals? How did the offspring map to the P genome? These details should be included in Supplementary Figure 1. Could the authors also provide information about the number of genes expressed at each stage in the F1 LP and PL samples in S Figure 2? How many genes went into the PCA? Many of these details are necessary to evaluate the F1 RNA-seq analyses.Generally, the authors need to report the statistics used in data processing more thoroughly. The authors need to report the statistics used to (1) process and evaluate the RNA-seq data and (2) determine the significance between the two morphs (Supplementary Figures 1 and 2).

(1) We clarified in the methods that F1 embryos are collected at the same stage (not absolute time) as the parental types. So the “16-cell” stage is comparable across planktotrophic, lecithotrophic and F1 offspring regardless of absolute time taken to reach that stage (which differs by ~3 hours- Figure 1).

Figure 2A details every time point collected for all crosses. As mentioned in the methods, we were unable to collect two timepoints for one set of crosses (LP) due to limited tissue. However, we still cover the full development time from “16 cell” through “swimming larvae” stages, which is the full larval development time.

(2) We appreciate the reviewer's concerns regarding the mapping to the reference genome. The S. benedicti genome is a largely complete and contiguous chromosome-length genome which we have now highlighted in the manuscript. However, the reference is only for the planktotrophic morph. So it is certainly possible that there could be mapping bias for lecithotrophic reads or F1 reads, as we point out in the discussion. While some bias is certainly possible, it is unlikely to be driving major differences in the results. We performed several tests to demonstrate this:

(1) We conducted two-sided T-tests of the mapping rates between all sample groups in our dataset (PP, LL, PL, LP) to determine if there were significant differences in mapping rates among the populations. No significant differences were found. The specific results of these statistical tests are included in the updated manuscript in supplementary figure 1 and are as follows:

**Author response table 1. sa3table1:** 

Comparison (Read Mapping Rates)	p-value of two sided T-test
PP versus LL	0.1118
PP versus PL	0.453
PL versus LP	0.418
LP versus LL	0.908

(2) In response to the comment about sequence level divergence affecting mapping rate, we estimated **pi** (nucleotide diversity within a population) and **dxy** (genomic divergence between two populations) based on the sampled transcriptomic data of our Planktotrophic and Lecithotrophic populations. We used PIXY (Korunes, K.L. and Samuk, K., 2021) with its standard settings to estimate these values, with variant call files in bcf format produced with bcftools - one for all planktotrophic samples and one for all lecithotrophic samples in our dataset. We found that across regions of the transcriptome, the difference in **pi** between Planktotrophs and Lecithotrophs was between 0.11% and 4.2%. Genomic divergence across the transcriptome is also relatively minor: estimates of **dxy** ranged from 0.0049 to 0.0076. Given that these estimates show relatively modest differences in nucleotide diversity and overall sequence divergence, we maintain that it is unlikely that they significantly impact the results described in this study. From what we have seen in the literature, these values are not outside of other population studies that are mapping to a species reference derived from one population.

We added the mapping rates of all samples in the Supplement (SFig. 1) as requested. We added the number of genes expressed at each stage in the Supplement (SFig. 2) as requested. We have also provided further details and figures (Fig 8B) on read mapping rates and statistics used in data processing, including those for F1 RNA-seq data.